# The serum vitamin D levels alleviate the influence of dietary inflammation on frailty: A cross-sectional analysis in the U.S. older adults

Yali Wang[1]☯, Ning Yan[2]☯, Yiling Luo[3], Kai Liu[1], Junru Wang[1], Jiahui Zhang[1], Xiaojun Ma[1], Jing Wang[1], Zhuoyuan Li[1], Liqun Wang ●[1]*

1 Department of Epidemiology and Statistics, School of Public Health at Ningxia Medical University, Yinchuan, China, 2 Heart Centre & Department of Cardiovascular Diseases, General Hospital of Ningxia Medical University, Yinchuan, China, 3 Department of Health Management Center, People's Hospital of Ningxia Hui Autonomous Region, Yinchuan, China

☯ These authors contributed equally to this work.
* wlq920@nxmu.edu.cn

## Abstract

Dietary inflammation (DI) and suitable vitamin D intake was associated with frailty. However, the possible mechanism that how DI affected frailty was still unclear. The current study aimed to explore the mediating association of vitamin D levels in the link between DI and frailty. A cross-sectional study of 1172 participants from the National Health and Nutrition Examination Survey (NHANES) database in 2011−2014 was conducted. DI was assessed by the Dietary Inflammation Index (DII), serum vitamin D level were assessed by serum 25-hydroxyvitamin D3 (25-(OH)D3), and frailty was assessed by 49-frailty index. The mediation package in R 4.3.3 was used to examine the mediating association of serum 25-(OH)D3 levels on the relationship between DII and frailty. The prevalence of frailty was 32.3%. DII was negatively correlated with 25-(OH)D3 ($r = −0.131$, $P < 0.001$), and positively related with frailty ($r = 0.131$, $P < 0.001$). 25-(OH)D3 was negatively associated with frailty ($r = −0.073$, $P = 0.013$). And the results showed that 25-(OH)D3 was a possible mediating association between DI and frailty, which explained 10.5% of the total effect (0.0004/0.0038). Improvements in DI and increased vitamin D levels may help alleviate frailty. People should pay more attention to the diet pattern of older adults.

## Background

With the development of an aging society, frailty is gradually becoming a public health problem threatening human health. Frailty is a state characterized by reduced physiological reserve and loss of resistance to stressors caused by accumulated age-related deficits [1]. There is an increasing agreement that indicators of frailty

**Data availability statement:** All relevant data are within the manuscript and its Supporting Information files.

**Funding:** This work was supported by Scientific Research Funding Project of Ningxia Medical University (grant number XT2022014).

**Competing interests:** The authors have declared that no competing interests exist.

**Abbreviations:** NHANES, National Health and Nutrition Examination Survey; US, United States; DII, Dietary Inflammation Index; FI, Frailty Index; 25-(OH)D3, 25-hydroxyvitamin D3; IL-1β, Interleukin-1 β; IL-4, Interleukin-4; IL-6, Interleukin-6; IL-10, Interleukin-10; TNF-α, Tumor Necrosis Factor-α; CRP, C-reactive protein; SBP, Systolic Blood Pressure; DBP, Diastolic Blood Pressure; PIR, Poverty Income Ratio; HDL, High-Density Lipoprotein; OR, Odds Ratio; IQR, Interquartile Range.

encompass age-related declines in lean body mass, strength, endurance, balance, walking ability, and low activity levels, and that frailty is defined in clinical by the presence of multiple contributing factors [2]. Frailty may also trigger and exacerbate other health issues [3], such as the elevated rates of mortality [4], hospitalization [5], and disability [6]. In the long term, frailty can be effectively treated or prevented.

Some researchers have suggested that better dietary quality or healthier eating patterns may reduce the risk of frailty in recent years [7]. Among them, dietary factors have attracted much attention as one of the controllable factors; it can affect frailty by regulating inflammatory factors and oxidative stress. Therefore, inflammation plays a vital role in the pathogenesis of debilitating diseases. To more fully and accurately quantify the inflammatory power of diet, Shivappa et al. created a literature-based, population-based Dietary Inflammation Index (DII) in 2014 [8]. The DII was calculated using an individual's record of food consumption and assessment of relevant nutrients. It is believed that a high DII score represents a pro-inflammatory diet, which is closely related to serum inflammatory markers [9,10]; however, a low score indicates anti-inflammatory properties [9,10]. A recent study on nutritional status and DII in older adults found that DII levels were higher in frail and malnourished individuals, and further to reduced their life quality [11]. The positive correlation between the DII and the risk of frailty in patients with osteoarthritis has also been confirmed [12]. Hence, understanding dietary inflammation's role may provide valuable insights for frailty prevention and treatment.

Vitamin D is a fat-soluble molecule referring to the different isoforms (D2 and D3). 25-hydroxyvitamin D3 (25 (OH) D3) is the main circulating form of vitamin D and widely used to assess vitamin D status [13]. A prospective cohort study found an association between low vitamin D levels and frailty and mortality among older adults [14]. In recent years, several studies have found that vitamin D has anti-inflammatory, antioxidant, and other biological functions [15–17], which make vitamin D a great potential for preventing and treating frailty in older population. Recent studies have identified a significant correlation between vitamin D and DII; specifically, vitamin D deficiency was associated with elevated DII levels [18]. This effect may be mediated through vitamin D's ability to regulate immune responses by modulating the balance between pro- and anti-inflammatory cytokines [19,20].

However, there still existed literature gaps of the potential mechanism between DII and frailty. Hence, in this study, we hypothesize that 1) DII is positively associated with frailty and negatively associated with vitamin D; 2) vitamin D is negatively correlated with frailty; 3) vitamin D may have a mediating association on the relationship between DII and frailty among U.S. older adults (as shown in Fig 1).

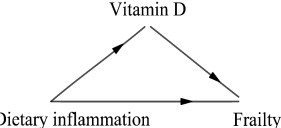

**Fig 1. Diagram for the mediation analysis.**

## Methods

### Study design and participants

The samples in this study were from the NHANES database (NHANES Questionnaires, Datasets, and Related Documentation). NHANES was conducted by the Center for Disease Control and Prevention (CDC), aimed to assess the health status of the U.S. non-institutionalized civilian population. The present analysis combined two NHANES cycles (NHANES 2011–2012 and 2013–2014). Owing to the complex, multistage, and probabilistic nature of the sampling design, the selected samples are representative of the non-institutionalized US civilian population. This combined sample included 1172 older adults aged 60 years or older. Among them, we excluded people with severe cardiovascular and cerebrovascular diseases, liver and kidney diseases, and people who could not provide biological samples [21,22], the details are shown in Fig 2. The NHANES protocols were approved by the National Center for Health Statistics (NCHS) Research Ethics Review Board (ERB). All participants provided written informed consent in accordance with federal regulations and institutional requirements. An individual investigator utilizing the publicly available NHANES data do not need to file the institution internal review board (IRB).

### Frailty

Frailty was assessed using frailty index (FI). The FI is a robust and concise instrument for assessing frailty in older adults, it performs well in distinguishing frailty states [2], and exhibits conceptual soundness and excellent content validity. A value ranging from 0 to 1 was assigned based on the severity of each deficit, allowing for the integration of continuous and categorical variables. Participants were divided into 4 subgroups: FI ≤ 0.1 (fit), 0.1 < FI ≤ 0.2 (vulnerable), 0.2 < FI ≤ 0.3 (mildly

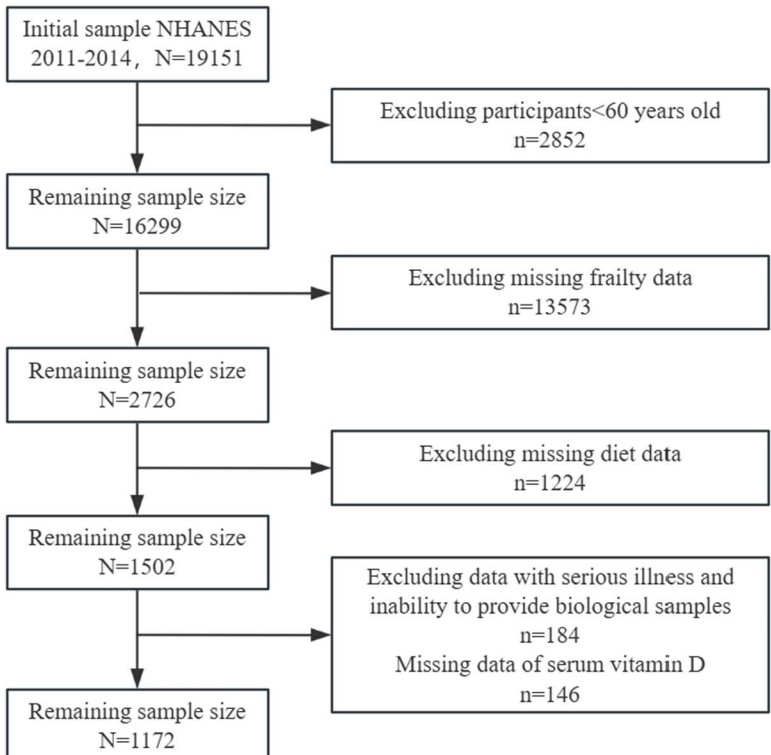

**Fig 2. Flowchart of the study design and participants.**

frail), and FI > 0.3 (moderately/severely frail) [14,23]. The frailty index comprised 49 deficits spanning multiple systems [24], including Cognition (Experience confusion/memory problems), Dependence(Managing money, Stooping, crouching, kneeling, and Lifting or carrying; etc., 15 items), Depressive Symptoms (Have little interest in doing things, Trouble sleeping or sleeping too much and Feeling tired or having little energy; etc., 7 items), Comorbidities (Arthritis, Thyroid problems and Chronic bronchitis; etc., 13 items), Hospital Utilization and Access to Care (Self-rated health, Health now compared with 1 year ago and Overnight hospital patient in past year; etc., 6 items), Laboratory Values (Glycohemoglobin, Red blood cell count and Hemoglobin; etc., 6 items) (S1 Table in S1 File). In this study, fit and vulnerable (i.e., FI ≤ 0.2) were combined into the no-frailty group, while mildly frail and moderately/severely frail (i.e., FI > 0.2) were combined into the frailty group.

### Dietary inflammatory

Dietary inflammatory (DI) was evaluated by DII. All NHANES participants were eligible to undergo two 24-hour dietary recall interviews. The first recall was conducted in person at a mobile examination center, while the second was carried out via telephone 3 to 10 days later. Dietary data were derived from the average of these two recalls.

The DII, developed by Shivappa through an extensive literature review, was utilized to assess the potential inflammatory levels associated with dietary components [8]. The DII quantifies the inflammatory potential of a diet by integrating its relationships with six key inflammatory biomarkers: IL-1β, IL-4, IL-6, IL-10, TNF-α, and CRP. The index estimates the inflammatory effects of the consumption of 45 nutrients, expressed as a z-score relative to a standardized global mean. The z-scores were calculated by subtracting the mean intake from a regionally representative database and dividing the result by the standard deviation of the respective nutrient. These z-scores were then transformed into proportions (ranging from 0 to 1), doubled, and centered by subtracting 1. The DII score for each participant was derived by summing all food parameter-specific DII values. In this study, 26 nutrients were used to compute the DII, including alcohol, vitamins B12 and B6, β-carotene, caffeine, carbohydrates, cholesterol, total fat, fiber, folic acid, iron, magnesium, zinc, selenium, MUFA, niacin, omega-3 and omega-6 fatty acids, protein, PUFA, riboflavin, saturated fat, thiamine, and vitamins A, C, and E. Importantly, the DII remained valid even when fewer than 30 nutrients were included in the calculation [8]. Participants were subsequently categorized into two dietary groups based on their DII scores: an anti-inflammatory diet (DII < 0) and a pro-inflammatory diet (DII ≥ 0) [25].

### Serum vitamin D levels

Participants' blood samples are collected by a certified phlebotomist at a Mobile Examination Center (MEC) under standardized protocols. The serum concentrations of 25-hydroxyvitamin D3 [25(OH)D3] (nmol/L) were quantified using high-performance liquid chromatography-tandem mass spectrometry (HPLC-MS/MS). The CDC LC-MS/MS method has better analytical specificity and sensitivity, and the analytical targets for imprecision(≤10%) and bias (≤5%) are fixed (25-Hydroxyvitamin D in Serum (cdc.gov)).

### Assessment of covariates

In this study, we selected several covariates associated with DI, frailty based on previously published research [25–27]. Covariates include numerical variables such as age (actual value), sitting time(min/d), diastolic blood pressure (DBP, mmHg), total cholesterol (mmol/L), high-density lipoprotein (HDL, mmol/L), vitamin B12 (pmol/L) and Energy ((kcal). Binary variables were sex (Female, Male), smoking (Yes or No), alcohol consumption (Yes or No), and physical activity (Yes or No); the multi categorical variables were ethnicity (Mexican American, Other Hispanic, Non-Hispanic White, Non-Hispanic Black, Other race), education(Less than 9th grade, 9–11th grade, High School graduate/GED or equivalent, Associate or associate degree, College graduate or higher), marriage(Married, Unmarried, Divorce, Other), and poverty income ratio (PIR) (<1.3, 1.3–3.49, ≥ 3.5).

 

## Statistical analysis

This study conducted statistical analyses using R version 4.3.3, with a *P*-value of less than 0.05 (two-tailed) considered statistically significant. Continuous but non-normal variables are represented as medians and quartiles (median [IQR]), and categorical variables are shown as frequency (%). Differences between continuous variables were assessed using Analysis of Variance (ANOVA), while the Chi-square test was employed for categorical variables. Logistic regression was used to analyze the relationship between participants' DI and frailty. Three sets of models were used: an unadjusted model (model 1), model 2 adjusting for sex, age, education, PIR and BMI, and model 3 adjusting for physical activity, DBP, total cholesterol, HDL and Energy. The results of logistic regression analysis were reported as odds ratio (OR) and 95%CI. Taking DI as independent variable, serum 25(OH)D3 levels as mediating variable, and frailty as dependent variable, the mediating association of serum 25(OH)D3 levels between DI and frailty was examined. The mediation package in R 4.3.3 was used to examine the mediating association of serum 25(OH)D3 levels between DI and frailty. Numerous studies have utilized publicly available data from the NHANES to investigate risk factors for various diseases. In these studies, some researchers have employed weighted analysis methods, while others have used unweighted approaches. Although NHANES employs complex sampling techniques to enhance the representativeness and generalizability of the survey results, discrepancies in conclusions may arise between weighted and unweighted analyses. In the present study, we conducted a sensitivity analysis using weighted logistic regression to reaffirm our findings. The weighting variable wtdrd2 were selected, and the combined weight was calculated as 1/2*wtdrd2. All sensitivity analyses were conducted using these weighted approaches.

## Results

### Demographic characteristics of participants

A total of 1172 participants were involved in the final study, of which 793 were non-frailty and 379 were frailty; males accounted for 46.59% of the sample, while females accounted for 53.41%. As shown in Table 1. Among all participants, the number of Pro-inflammatory diet (DII ≥ 0) accounted for 76.28% of the total sample, and 309 people in the sample with high DII were frailty. Significant differences were observed between the frailty and non-frailty groups across demographic characteristics (age, sex, ethnicity, marriage, PIR), clinical parameters (physical activity, sedentary time, total cholesterol, DBP, HDL, height, weight, BMI, hypertension, diabetes), and nutritional profiles (energy, protein, carbohydrates, and DII) (all *P* < 0.05).

### Correlation between DI, Serum vitamin D and frailty

Through the normality test, DI was normally distributed, with Mean ±SD (1.46 ± 1.92). 25-(OH)D3 and frailty index showed skewed distribution, and median[IQR] were 73.61 [53.41,91.47] and 0.16 [0.12, respectively. 0.23]. DI was negatively correlated with 25-(OH)D3 (r = −0.14, *P* < 0.001), and positively correlated with frailty (r = 0.14, *P* < 0.001). 25-(OH)D3 was negatively correlated with frailty (r = −0.07, *P* = 0.013), and the results were shown in Table 2.

### Relationship between DI, Serum vitamin D and frailty

Table 3 shows the results of the logistic regression of DII and frailty by three models. After adjusting for potential confounders (in Model 3), the results showed a significant association between DII and frailty, when DII was a continuous variable (OR = 1.08, 95%CI [1.01~1.16], *P* = 0.044). When DII was categorized into anti-inflammatory and pro-inflammatory diets, with the anti-inflammatory diet serving as the reference group, no significant association was observed between the pro-inflammatory dietary pattern and frailty (OR = 1.13, 95% CI [0.80~1.62], *P* = 0.471). When DII was modeled as quantiles, it was found that after adjusting for confounding factors (in Model 3), compared to quartile Q1, the odds ratios (OR) for Q2, Q3, and Q4 were all greater than 1, but showed no statistical significance (*P* > 0.05). These results suggest that,

**Table 1. Demographic characteristics of participants (n = 1172).**

| Subgroups | Total sample | Frailty | | P |
|---|---|---|---|---|
| | | Frailty | Non-frailty | |
| **Total sample (n (%))** | 1172 | 379 (32.33) | 793 (67.66) | |
| **DII (Mean±SD)** | 1.46±1.92 | 1.80±1.81 | 1.29±1.95 | <0.001 |
| **DII** Anti-inflammatory Pro-inflammatory | 278(23.72) 894(76.28) | 70(18.47) 309(81.53) | 208(26.23) 585(73.77) | 0.003 |
| **Age (median (IQR))** | 68.00[64.00,74.00] | 70.00 [64.00, 77.00] | 68.00 [63.00, 73.00] | <0.001 |
| **Sex (n (%))** | | | | <0.003 |
| Male | 546 (46.59) | 153(40.37) | 393(49.56) | |
| Female | 626 (53.41) | 226(59.63) | 400(50.44) | |
| **Ethnicity (n (%))** | | | | <0.021 |
| Mexican American | 72 (6.14) | 20 (5.28) | 52 (6.56) | |
| Other Hispanic | 102 (8.70) | 31 (8.18) | 71 (8.95) | |
| Non-Hispanic White Non-Hispanic Black Others | 651 (55.55) 263 (22.44) 84 (7.17) | 216 (56.99) 97 (25.59) 15 (3.96) | 435 (54.85) 166 (20.93) 69 (8.70) | |
| **Education (n (%))** | | | | <0.073 |
| Less than 9th grade | 77 (6.57) | 31 (8.18) | 46 (5.80) | |
| 9-11th grade | 145 (12.37) | 51 (13.46) | 94 (11.85) | |
| High School graduate/GED or equivalent | 267 (22.78) | 97(25.59) | 170(21.44) | |
| Associate or associate degree | 355 (30.29%) | 110 (29.02%) | 245 (30.90%) | |
| College graduate or higher | 328 (27.99%) | 90 (23.75%) | 238 (30.01%) | |
| **Marriage (n (%))** | | | | 0.002 |
| Be married | 708 (60.41) | 212 (55.94) | 496 (62.55) | |
| Unmarried | 52 (4.44) | 14 (3.69) | 38 (4.79) | |
| Divorce | 168 (14.33) | 61 (16.09) | 107 (13.49) | |
| Others | 244 (20.81) | 92 (24.27) | 152 (19.16) | |
| **Height (Mean±SD (cm))** | 166 (158, 173) | 164±10 | 167±10 | <0.001 |
| **Weight (Mean±SD (kg))** | 78 (67, 91) | 83±21 | 78±17 | <0.001 |
| **BMI(Mean±SD)** | 27.9 (24.8, 32.0) | 30.8±7.1 | 28.0±5.0 | <0.001 |
| **PIR (n (%))** | | | | <0.001 |
| <1.3 | 235 (20.05) | 92 (24.27) | 143 (18.03) | |
| 1.3-3.49 | 474 (40.44) | 177 (46.70) | 297 (37.45) | |
| >=3.5 | 463 (27.99) | 110 (29.02) | 353 (44.51) | |
| **Smoking (n (%))** | | | | <0.113 |
| Yes | 573(48,89) | 198 (52.24) | 375 (47.29) | |
| No | 599(51.11) | 181 (47.76) | 418 (52.71) | |
| **Alcohol consumption (n (%))** | | | | <0.180 |
| Yes | 822 (70.14) | 256(67.55) | 566 (71.37) | |
| No | 350 (29.86) | 123(32.45) | 227 (28.63) | |
| **Physical activity (n (%))** | | | | <0.001 |
| Inactive | 553 (47.18) | 140 (36.94) | 413(52.08) | |
| Active | 619 (52.82) | 239 (63.06) | 380(47.92) | |
| **Hypertension** | | | | <0.001 |
| Yes | 778 (66%) | 290(76.52%) | 488(61.54%) | |
| No | 394 (34%) | 89(23.48%) | 305(38.46%) | |

*(Continued)*

**Table 1.** (Continued)

| Subgroups | Total sample | Frailty | | P |
|---|---|---|---|---|
| | | Frailty | Non-frailty | |
| **Diabetes** | | | | <0.001 |
| Yes | 260 (22%) | 126(33.25%) | 134(16.90%) | |
| No | 912(77.81%) | 253(66.75%) | 659(83.10%) | |
| **Total fat(Mean±SD (gm))** | 67 (44, 93) | 70±38 | 73±37 | 0.116 |
| **Total saturated fatty Acids (Mean±SD (gm))** | 20 (13, 29) | 22±13 | 23±13 | 0.190 |
| **Total monounsaturated fatty acids (Mean±SD (gm))** | 23 (15, 33) | 25±15 | 26±14 | 0.153 |
| **Total polyunsaturated fatty acids (Mean±SD (gm))** | 15 (10, 23) | 17±11 | 18±11 | 0.211 |
| **Sedentary time (median (IQR))** | 360.00(240.00,480.00) | 420.00(300.00,540.00) | 360.00(240.00,480.00) | <0.001 |
| **SBP (median (IQR))** | 130.67(120.00,141.33) | 130.67(118.00,140.33) | 130.67(120.67,142.00) | <0.458 |
| **DBP (median (IQR))** | 68.67 (61.33, 75.33) | 66.67 (58.00, 74.00) | 70.00 (62.67, 76.00) | <0.001 |
| **Totalcholesterol(median(IQR))** | 4.84 (4.14, 5.64) | 4.68 (4.00, 5.46) | 4.91 (4.19, 5.66) | 0.008 |
| **HDL (median (IQR))** | 1.37 (1.14, 1.66) | 1.34 (1.11, 1.63) | 1.37 (1.16, 1.68) | 0.096 |
| **Vitamin B12 (median (IQR))** | 416.20(292.20,11.10) | 419.20(298.20,607.05) | 415.50(289.65,612.9) | 0.918 |
| **Energy (Mean±SD (kcal))** | 1742 (1299, 2253) | 1,740±694 | 1,887±780 | 0.004 |
| **Protein (Mean±SD (gm)** | 67 (49, 90) | 68±34 | 74±33 | 0.001 |
| **Carbohydrate (Mean±SD (gm)** | 207 (155, 273) | 208±80 | 229±103 | 0.014 |

**Table 2.** Correlation matrix (n=1172).

| | M(P25, P75)/X̄±s | DII | 25-(OH)D3 | FI |
|---|---|---|---|---|
| **DII** | 1.45±1.92 | 1 | | |
| **25-(OH)D3** | 73.61(53.42,91.47) | 0.131** | 1 | |
| **FI** | 0.18±0.09 | −0.131** | −0.073* | 1 |

**p<0.001, *p<0.05.

after adjusting for confounding factors, there may be a positive association between DII and frailty. There was no statistical significance in the interaction between DI and 25-(OH)D3 ($P>0.05$), Therefore, we suggest that 25-(OH)D3 may play a mediating role.

## Relationship between DII and frailty stratified by age and sex

Table 4 displays the correlation between DII and frailty in various subgroups sex. We observed that after adjusting for demographic and lifestyle factors (In Model 3), DII were positively associated with frailty in males, and the difference was statistically significant (Males: OR=1.19, 95% CI [1.06–1.33], $P=0.003$). DII was also positively correlated with frailty in female but the difference was not statistically significant (Female: OR=1.03, 95%CI [0.93–1.14], $P=0.570$). Table 4 shows the relationship between DII and frailty in subgroups of different ages. We observed that in the 60–70 age group, after adjusting for potential confounders (In Model 3), there was a significant association between DI and frailty (OR=1.12, 95%CI [1.01~1.24], $P=0.035$). In the 70–80 age group, we observed a statistically significant positive correlation between dietary inflammatory index and frailty in older adults in unadjusted model 1(OR=1.10, 95%CI [1.01~1.21], $P=0.047$). In the final model (In Model 3), after controlling for confounders, DII was also positively correlated with frailty in older adults, but the difference was not statistically significant (OR=1.08, 95%CI [0.98~1.20], $P=0.129$).

**Table 3. Logistic regression analysis of DII, 25-(OH)D3 and frailty.**

| Subgroup | Model 1 | | | Model 2 | | | Model 3 | | |
|---|---|---|---|---|---|---|---|---|---|
| | OR | 95%CI | P | OR | 95%CI | P | OR | 95%CI | P |
| DII | 1.15 | (1.08~1.23) | <0.001 | 1.09 | (1.01~1.17) | 0.018 | 1.08 | (1.01~1.16) | 0.044 |
| DII | | | | | | | | | |
| <0 | Ref | 1 | | Ref | 1 | | Ref | 1 | |
| >=0 | 1.57 | (1.16~2.13) | 0.004 | 1.25 | (0.90~1.74) | 0.175 | 1.13 | (0.80~1.62) | <0.471 |
| DII Quantile | | | | | | | | | |
| Q1 | Ref. | 1 | | Ref. | 1 | | Ref. | 1 | |
| Q2 | 1.38 | (0.96~1.99) | 0.080 | 1.12 | (0.76~1.65) | 0.539 | 1.12 | (0.72~1.60) | 0.548 |
| Q3 | 1.57 | (1.09~2.24) | 0.014 | 1.32 | (0.90~1.95) | 0.150 | 1.36 | (0.83~1.90) | 0.117 |
| Q4 | 2.04 | (1.43~2.91) | <0.001 | 1.50 | (1.02~2.20) | 0.036 | 1.49 | (1.01~2.19) | 0.041 |
| 25-(OH)D3 | 1.00 | (0.99~1.00) | <0.132 | 0.99 | (0.99~1.00) | 0.744 | 1.00 | (0.99~1.00) | 0.932 |
| DII×25-(OH)D3 | 1.00 | (1.00~1.00) | 0.004 | 0.99 | (0.99~1.00) | 0.129 | 0.99 | (0.99~1.00) | 0.410 |

Model 1: unadjusted.

Model 2: adjusted for age, sex, education, PIR and BMI.

Model 3: Model 2+Physical activity, DBP, Total cholesterol, HDL and Energy.

**Table 4. The association between DII and frailty was observed by subgroup analysis.**

| Subgroup | Model 1 | | | Model 2 | | | Model 3 | | |
|---|---|---|---|---|---|---|---|---|---|
| | OR | 95%CI | P | OR | 95%CI | P | OR | 95%CI | P |
| Sex | | | | | | | | | |
| Male | 1.22 | (1.10~1.35) | <0.001 | 1.20 | (1.07~1.33) | 0.001 | 1.19 | (1.06~1.33) | 0.003 |
| Female | 1.08 | (0.98~1.18) | 0.117 | 1.03 | (0.94~1.14) | 0.527 | 1.03 | (0.93~1.14) | 0.570 |
| Age | | | | | | | | | |
| [60~70) | 1.20 | (1.10~1.32) | <0.001 | 1.15 | (1.04~1.28) | 0.005 | 1.12 | (1.01~1.24) | 0.035 |
| [70~80] | 1.10 | (1.01~1.21) | 0.047 | 1.07 | (0.97~1.19) | 0.173 | 1.08 | (0.98~1.20) | 0.129 |

Model 1: unadjusted.

Model 2: adjusted for age, sex, education, PIR and BMI.

Model 3: Model 2+Physical activity, DBP, Total cholesterol, HDL and Energy.

## Mediator role of Serum vitamin D in DI and frailty

As shown in Table 5, after controlling for covariates, serum 25-(OH)D3 has a significant mediation association on the relationship between DII and frailty. The results showed that both the direct effect (P<0.001) and the indirect effect (P<0.001) were significant. The mediation association explained 10.5% (0.0004/0.0038) of the total variance. Fig 3 visualizes the results of the mediating association of 25-(OH)D3.

## Sensitivity analysis

Similarly, the sensitivity analysis using weighted logistic regression indicated that when DII was treated as a continuous variable, a one-unit increase in DII was associated with a 9% increase in the risk of frailty in Model 3 (OR = 1.09, 95% CI: 1.01–1.18, P=0.019). When DII was dichotomized (pro-inflammatory vs. anti-inflammatory), individuals with a pro-inflammatory diet had an 18% higher risk of frailty compared to those with an anti-inflammatory diet in Model 3 (OR = 1.18, 95% CI: 0.89–1.57, P=0.216), although this association was no longer statistically significant. When DII was

**Table 5. The mediating association of 25-(OH)D3 on the relationship between DII and frailty among older adults.**

| | β | SE | p | Bias-corrected 95%CI | |
|---|---|---|---|---|---|
| | | | | Lower | Upper |
| Total effect | 0.0038 | 0.0014 | 0.006 | 0.0011 | 0.0065 |
| Direct Effects | 0.0034 | 0.0004 | 0.013 | 0.0007 | 0.0062 |
| Indirect Effects | 0.0004 | 0.0002 | 0.0047 | 0.0001 | 0.0008 |

After controlling for age, sex, ethnicity, PIR, education, smoking, alcohol, sitting time, HDL, vitaminB12.

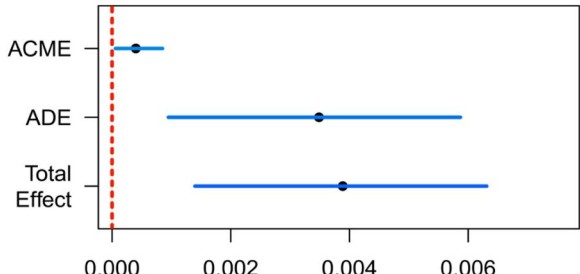

**Fig 3. The mediation association of 25-(OH)D3 on DI and frailty.**

categorized into quartiles, individuals in the highest quartile (Q4) had a 62% higher risk of frailty compared to those in the lowest quartile (Q1) in Model 3 (OR = 1.62, 95% CI: 1.16–2.26, *P* = 0.006) (Table 6). The weighted mediation association accounted for 6.8% of the total variance (0.0129/0.1886) (Table 7). Subgroup analysis using the weighted approach are detailed in the supplementary material (S2 Table in S1 File).

**Table 6. Weighted Logistic regression analysis of DII, 25-(OH)D3 and frailty.**

| Subgroup | Model 1 | | | Model 2 | | | Model 3 | | |
|---|---|---|---|---|---|---|---|---|---|
| | OR | 95%CI | *P* | OR | 95%CI | *P* | OR | 95%CI | *P* |
| DII | 1.17 | (1.11~1.24) | <0.001 | 1.11 | (1.04~1.18) | 0.001 | 1.09 | (1.01~1.18) | 0.019 |
| DII | | | | | | | | | |
| <0 | Ref | 1 | | Ref | 1 | | Ref | 1 | |
| >=0 | 1.65 | (1.33~2.06) | 0.001 | 1.35 | (1.03~1.75) | 0.027 | 1.18 | (0.89~1.57) | 0.216 |
| DII Quartile | | | | | | | | | |
| Q1 | Ref. | 1 | | Ref. | 1 | | Ref. | 1 | |
| Q2 | 1.44 | (1.05~1.96) | 0.023 | 1.16 | (0.81~1.67) | 0.374 | 1.13 | (0.78~1.64) | 0.473 |
| Q3 | 1.59 | (1.08~2.33) | 0.019 | 1.40 | (0.94~2.07) | 0.090 | 1.39 | (0.94~2.06) | 0.088 |
| Q4 | 2.19 | (1.66~2.89) | <0.001 | 1.68 | (1.22~2.32) | 0.003 | 1.62 | (1.16~2.26) | 0.006 |
| 25-(OH)D3 | 0.99 | (0.99~1.00) | 0.145 | 0.99 | (0.99~1.00) | 0.408 | 0.99 | (0.99~1.00) | 0.932 |
| DII×25(OH)D3 | – | – | <0.001 | _ | _ | 0.027 | 1.00 | (0.99~1.00) | 0.182 |

Model 1: unadjusted.

Model 2: adjusted for age, sex, education, PIR and BMI.

Model 3: Model 2 + Physical activity, DBP, Total cholesterol, HDL and Energy.

**Table 7. Mediating association between DII and frailty in older adults after weighted 25-(OH)D3.**

| | β | SE | p | Bias-corrected 95%CI | |
|---|---|---|---|---|---|
| | | | | Lower | Upper |
| Total effect | 0.1886 | 0.0228 | 0.0001 | 0.1349 | 0.23 |
| Direct Effects | 0.1757 | 0.0225 | 0.0001 | 0.1223 | 0.22 |
| Indirect Effects | 0.0129 | 0.0028 | 0.0001 | 0.0092 | 0.02 |

After controlling for age, sex, ethnicity, PIR, education, smoking, alcohol, sitting time, HDL, vitaminB12.

## Discussion

Based on nationwide representative sample data, the aim of this study was to investigate the relationship between DI, serum vitamin D, and frailty in old age. The results showed DII is positively associated with frailty and negatively associated with vitamin D in old adults. In the relationship between DI and frailty, serum vitamin D levels played a mediator role, accounting for 10.5% of the total. In addition, men showed higher frailty compared to women, and in the age group of 60–70, the higher the DII, the more likely they are to be frailty. To our knowledge, this is the first study to examine the mediating association of serum vitamin D levels between DI and frailty.

This study reported a significant positive correlation between DI and frailty, consistent with previous findings that DI elevates frailty risk [28]. In a survey of people with depression, there existed association between a pro-inflammatory diet and the onset of frailty [29], and pro-inflammatory diets were associated with an increased risk of frailty. The underlying molecular mechanism may involve pro-inflammatory cytokines exerting a direct influence on frailty by promoting protein degradation or an indirect effect via key metabolic pathways [30]. Our study found a negative association between serum 25-(OH)D3 levels and frailty. This finding was consistent with previous research, which also confirmed a significant association between serum vitamin D levels and frailty [31]. Moreover, studies have documented that low levels of 25(OH)D are linked to impaired balance [32], an increased risk of falls [33], a higher incidence of fractures [34], and persistent pain [35]. These factors can contribute to a sedentary lifestyle and immobility, ultimately leading to increased frailty [36,37]. Deficiency of vitamin D as a prohormone [38] may lead to a series of physiological dysfunction by reducing serum calcium concentration [39], which in the long run can impair the body's ability to cope with environmental stress and disrupt homeostasis. Supporting this premise, a study conducted on murine models reported that high-dose vitamin D supplementation mitigated frailty and improved balance [40]. Therefore, maintaining appropriate vitamin D levels is essential for maintaining musculoskeletal health and physiological homeostasis.

In this study, serum vitamin D levels were found to mediate between DI and frailty. The most crucial reason for this may be that vitamin D not only affects the homeostasis of mineral metabolism but also possesses anti-inflammatory and antioxidant properties [41,42]. It plays a role in anti-angiogenesis by regulating various aspects of the cell cycle, including cell division, proliferation, and apoptosis [43]. Specifically, Vitamin D directly regulates the inflammatory microenvironment by inhibiting the expression of pro-inflammatory cytokines (e.g., *TNF-α, IL-6*) and upregulating anti-inflammatory factors (e.g., *IL-10*) [44,45]. We found that individuals with lower serum vitamin D levels had a significantly higher risk of frailty in the face of higher DII, it is suggested that diet-induced chronic inflammation may exacerbate the debilitating process through vitamin D deficiency. This is consistent with previous studies that the vitamin D receptor (VDR) is widely expressed in immune cells and its deficiency impairs negative feedback regulation of inflammatory signaling [46], leads to muscle atrophy, loss of bone density and impaired nerve function, eventually manifesting as a debilitating syndrome [47,48]. A possible mechanism for this is that chronic low-grade inflammation leads to degeneration of the bone-muscle system through activation of osteoclasts and inhibition of muscle protein synthesis [49,50], while vitamin D disrupts this vicious cycle through its dual actions of regulating calcium-phosphorus metabolism and exerting anti-inflammatory effects [51].

The study acknowledges the presence of several strengths and limitations. Broadening the knowledge regarding the underlying mechanisms in the relationship between DII and frailty of older adults, particularly the mediating role of serum vitamin D in the relationship between DII and frailty was explored. However, several limitations of the study must also be clarified. Firstly, given the type of study in which cross-sectional studies are conducted, causation cannot be determined. In the future, further prospective studies are needed to determine the exact relationship between DII and frailty. Secondly, the diet inflammation index was measured through 24-hour recall interviews. Therefore, the diet at the time of the interview may not reflect long-term habitual eating patterns. However, previous work has demonstrated the effectiveness and practicality of this approach to assess the link between diet and health outcomes [52–54]. In addition, the diet of older adults was found to be relatively stable for at least five to seven years [55], suggesting that 24-hour recall may be an accurate reflection of eating over a longer period of time. Finally, NHANES is a representative survey of the US, and its conclusions may not be generalized to other countries, and should be applied with caution.

## Conclusions

The results of this study indicate that reducing DI and increasing serum vitamin D levels may help alleviate frailty in older people. This finding provides important implications for public health policy; that is, when formulating measures to prevent and intervene in aging faltering, diet quality should be considered comprehensively, and the health level of the older people should be improved by improving their diet habits. In the future, further prospective studies are needed to determine the exact relationship between DII and frailty.

## Supporting information

**S1 File.** S1 Table Variables in the 49-Item Frailty Index and Their Respective Scorings. S2 Table The relationship between DII and frailty was observed by subgroup analysis after weighting. S1 Data.
(ZIP)

## Author contributions

**Data curation:** Yali Wang, Ning Yan, Yiling Luo, Kai Liu, Junru Wang, Xiaojun Ma, Jing Wang, Liqun Wang.

**Formal analysis:** Yali Wang, Ning Yan, Yiling Luo, Kai Liu, Junru Wang, Jiahui Zhang, Xiaojun Ma, Zhuoyuan Li.

**Software:** Yali Wang, Ning Yan, Yiling Luo, Kai Liu, Jiahui Zhang, Xiaojun Ma, Zhuoyuan Li.

**Visualization:** Jing Wang, Liqun Wang.

**Writing – original draft:** Yali Wang, Liqun Wang.

**Writing – review & editing:** Yali Wang, Ning Yan, Liqun Wang.

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
