## [Decision Letter · Decision Letter 0]

PONE-D-24-52566The serum vitamin D levels alleviate the influence of dietary inflammation on frailty: a cross-sectional analysis in the U.S. older adultsPLOS ONE

Dear Dr. Wang

Thank you for submitting your manuscript to PLOS ONE. After careful consideration, we feel that it has merit but does not fully meet PLOS ONE’s publication criteria as it currently stands. Therefore, we invite you to submit a revised version of the manuscript that addresses the points raised during the review process.

Please go over the reviewers comments and address all the points they raised. 

We look forward to receiving your revised manuscript.

Kind regards,

Everson Nunes, Ph.D.

Academic Editor

PLOS ONE

Journal Requirements:

4. In this instance it seems there may be acceptable restrictions in place that prevent the public sharing of your minimal data. However, in line with our goal of ensuring long-term data availability to all interested researchers, PLOS’ Data Policy states that authors cannot be the sole named individuals responsible for ensuring data access (http://journals.plos.org/plosone/s/data-availability#loc-acceptable-data-sharing-methods ).

Reviewers' comments:

Reviewer's Responses to Questions

**Comments to the Author**

1. Is the manuscript technically sound, and do the data support the conclusions?

Reviewer #1: Yes

Reviewer #2: Yes

2. Has the statistical analysis been performed appropriately and rigorously? 

Reviewer #1: Yes

Reviewer #2: No

3. Have the authors made all data underlying the findings in their manuscript fully available?

Reviewer #1: No

Reviewer #2: Yes

4. Is the manuscript presented in an intelligible fashion and written in standard English?

Reviewer #1: Yes

Reviewer #2: Yes

5. Review Comments to the Author

Reviewer #1: In this paper, the authors conducted a cross-sectional analysis of older adults in the United States to explore the mediating role of vitamin D in the relationship between dietary inflammation and frailty. The findings have major implications for public health and nutrition in the aging population. lt is an interesting work, and the following issues need to be addressed before the article is published:

revisions.1. The introduction seems to be long and could be further shortened so that readers can grasp the core content of the study more quickly. For example, the third and fourth paragraphs could be integrated into one paragraph.

The Discussion section should refer to the latest literature in the past 3-5 years to explore in more detail how serum vitamin D levels mediate the relationship between DII and frailty and how these findings relate to existing theories of immune and inflammatory responses.

revisions.3. In lines 154-155, "Ratio of family income to poverty (RIP)" should be changed to "Ratio of family income to poverty (PIR)". Please see the full text.

revisions.4. Please add a flow chart of the study participants to this article.

revisions.5. What is the basis for the adjustment of confounders included in this article? In addition, it is recommended that baseline data such as BMI, diabetes, and hypertension should be included.

revisions.6.In the future, further prospective studies are needed to determine the exact relationship between DII and frailty.

Reviewer #2: The manuscript titled "The serum vitamin D levels alleviate the influence of dietary inflammation on frailty: a cross-sectional analysis in the U.S. older adults" presents an important and relevant contribution to the field of nutrition and aging. The study explores the mediating effect of vitamin D levels in the relationship between dietary inflammation and frailty, which is a timely and significant topic given the growing interest in strategies to mitigate frailty in aging populations. However, before this promising study can be accepted for publication, several revisions are required. These modifications will help enhance the clarity, rigor, and overall quality of the manuscript. Below are specific suggestions for revision.

In several instances throughout the manuscript, the authors use the term effect, such as in the objective stated in the abstract: "The current study aimed to explore the mediating effect of vitamin D levels in the link between DI and frailty." However, since this is a cross-sectional study, causality cannot be determined. Therefore, it would be more appropriate to replace effect with relationship or association to better reflect the study design and its limitations.

Methods

I suggest that the authors provide a more detailed explanation of the parameters used for frailty classification. Clarifying the specific criteria or thresholds applied in the frailty index would improve the transparency and reproducibility of the study.

I suggest that the authors specify which sampling weight was used to ensure the representativeness of the data, given that the study is based on a complex sampling design. Including this information would enhance the clarity and validity of the statistical approach.

In the Statistical Analysis section, I suggest including additional covariates that are relevant confounders in the relationship between vitamin D, frailty, and dietary inflammation. Specifically, I recommend incorporating an adiposity parameter, such as weight or preferably BMI, as well as a dietary intake parameter, such as total caloric intake. These variables would strengthen the analysis by accounting for important factors that may influence the observed associations.

Table 1 (Participant Characteristics) is missing important parameters that would allow for a better understanding of the sample. I suggest including weight, height, BMI, and dietary intake variables, such as total calorie intake, protein, carbohydrates, total fats, and fat subtypes. These additions would provide a more detailed characterization of the study population and improve the overall interpretation of the findings.

---

## [Author Response · Author response to Decision Letter 1]

13 Mar 2025

Dear Editors and reviewers,

We are grateful for the valuable comments and suggestions given by the reviewers and editors, and found them very useful in improving our manuscript, the submission ID is PONE-D-24-52566.

We give our responses to the reviewers in this letter. We hope that after our replies, our manuscript will be considered by PLOS ONE.

Q1: Please ensure that your manuscript meets PLOS ONE' s style requirements, including those for file naming.

Response: Thank you for your careful review. We have checked the manuscript item by item to ensure full compliance with the PLOS ONE format specification.

Q2: We note that the grant information you provided in the ‘Funding Information’ and ‘Financial Disclosure’ sections do not match.

Response: Thank you for your careful review. Due to an error during the online submission process, the funding information was incorrectly stated as "The authors received no specific funding for this work." As a result, the 'Funding Information' and 'Financial Disclosure' sections do not match. This work was supported by Scientific Research Funding Project of Ningxia Medical University (grant number XT2022014). We kindly request your assistance in updating the online submission form to reflect the correct funding information provided above.

Q3: Thank you for stating the following financial disclosure:

d) If you did not receive any funding for this study, please state: “The authors received no specific funding for this work.” Please include your amended statements within your cover letter; we will change the online submission form on your behalf.

Response: Thank you for your feedback. We would like to clarify the funding information for our study: This work was supported by Scientific Research Funding Project of Ningxia Medical University (grant number XT2022014). Due to an error during the online submission process, the funding information was incorrectly stated as "The authors received no specific funding for this work." We kindly request your assistance in updating the online submission form to reflect the correct funding information provided above.

Q4: In this instance it seems there may be acceptable restrictions in place that prevent the public sharing of your minimal data. However, in line with our goal of ensuring long-term data availability to all interested researchers, PLOS’ Data Policy states that authors cannot be the sole named individuals responsible for ensuring data access (http://journals.plos.org/plosone/s/data-availability#loc-acceptable-data-sharing-methods).

Response: Thank you for your valuable comments and suggestions. We have updated the Data Availability Statement in the article to make it clear that the study data is provided as supplementary materials (Data S1), and we have updated the Ethics Statement section to ensure that all research activities are ethically compliant and that appropriate ethical approvals have been obtained.

Q5: PLOS requires an ORCID iD for the corresponding author in Editorial Manager on papers submitted after December 6th, 2016. Please ensure that you have an ORCID iD and that it is validated in Editorial Manager. To do this, go to ‘Update my Information’ (in the upper left-hand corner of the main menu), and click on the Fetch/Validate link next to the ORCID field. This will take you to the ORCID site and allow you to create a new iD or authenticate a pre-existing iD in Editorial Manager.

Response: Thank you for your comment regarding the ORCID iD requirement. Corresponding author have already obtained an ORCID iD and validated it in Editorial Manager. The ORCID iD is [0000-0002-4316-3489], and it has been successfully authenticated in the system.

Reviewer #1

Q1: The introduction seems to be long and could be further shortened so that readers can grasp the core content of the study more quickly. For example, the third and fourth paragraphs could be integrated into one paragraph.

Response: Thank you very much for your professional advice. The introduction section has been streamlined to allow readers to access the core findings of this study more promptly and efficiently. We have made revisions and marked in red in the introduction part.

Q2: The Discussion section should refer to the latest literature in the past 3-5 years to explore in more detail how serum vitamin D levels mediate the relationship between DII and frailty and how these findings relate to existing theories of immune and inflammatory responses.

Response: We have updated the relevant literature, referencing the latest studies from the past 3-5 years. Additionally, we have provided a more detailed exploration of how serum vitamin D levels mediate the relationship between the DII and frailty, as well as how these findings relate to existing theories of immune and inflammatory responses. We have also elaborated on the potential mechanisms involved. All revisions were marked in red in the discussion part, can be seen in the lines 298-317, pages 14-15.

Q3: In lines 154-155, "Ratio of family income to poverty (RIP)" should be changed to "Ratio of family income to poverty (PIR)". Please see the full text.

Response: Many thanks. We have revised the “Ratio of family income to poverty (PIR)” as“Ratio of family income to poverty (RIP)”.

Q4: Please add a flow chart of the study participants to this article.

Response: Thank you for your valuable suggestion to improve the clarity of participant flow. We have added the participant flow chart (Figure 1) in the Study Design and Participants section, which systematically presents the complete process of sample selection and final inclusion in the analysis.

Q5: What is the basis for the adjustment of confounders included in this article? In addition, it is recommended that baseline data such as BMI, diabetes, and hypertension should be included.

Response: Thank you very much for your professional advice. I have added the variables of BMI, height, weight, energy intake, diabetes and hypertension in the baseline Characteristics Table. For details, see Table 1 (Participant Characteristics). Secondly, in this study, we selected several covariates associated with dietary inflammation and frailty based on previously published research to adjust for potential confounders. The literature is as follows:

[1] Ma R, Zhou X, Zhang G, Wu H, Lu Y, Liu F, Chang Y, Ding Y. Association between composite dietary antioxidant index and coronary heart disease among US adults: a cross-sectional analysis. BMC Public Health. 2023 Dec 5;23(1):2426. doi: 10.1186/s12889-023-17373-1. PMID: 38053099; PMCID: PMC10699074.

[2] Sun M, Wang L, Wang X, Tong L, Fang J, Wang Y, Yang Y, Li B. Interaction between sleep quality and dietary inflammation on frailty: NHANES 2005-2008. Food Funct. 2023 Jan 23;14(2):1003-1010. doi: 10.1039/d2fo01832b. PMID: 36546877.

[3] Mao Y, Weng J, Xie Q, Wu L, Xuan Y, Zhang J, Han J. Association between dietary inflammatory index and Stroke in the US population: evidence from NHANES 1999-2018. BMC Public Health. 2024 Jan 2;24(1):50. doi: 10.1186/s12889-023-17556-w. PMID: 38166986; PMCID: PMC10763382.

Q6: In the future, further prospective studies are needed to determine the exact relationship between DII and frailty.

Response: We fully agree with your point regarding the need for further prospective studies to determine the exact relationship between DII and frailty. This is indeed a crucial area for future research.

We have incorporated your suggestion into the limitation part, and highlighted in red, and the revised text is transcribed below:

“The study acknowledges the presence of several strengths and limitations. Broadening the knowledge regarding the underlying mechanisms in the relationship between DII and frailty of older adults, particularly the mediating role of serum vitamin D in the relationship between DII and frailty was explored. However, several limitations of the study must also be clarified. Firstly, given the type of study in which cross-sectional studies are conducted, causation cannot be determined. In the future, further prospective studies are needed to determine the exact relationship between DII and frailty. Secondly…

Reviewer #2:

Q1: In several instances throughout the manuscript, the authors use the term effect, such as in the objective stated in the abstract: "The current study aimed to explore the mediating effect of vitamin D levels in the link between DI and frailty." However, since this is a cross-sectional study, causality cannot be determined. Therefore, it would be more appropriate to replace effect with relationship or association to better reflect the study design and its limitations.

Response: Thank you very much for your insightful comments. We have carefully revised the manuscript to replace “effect” with “association” throughout. And highlighted in red in the manuscript.

Q2: I suggest that the authors provide a more detailed explanation of the parameters used for frailty classification. Clarifying the specific criteria or thresholds applied in the frailty index would improve the transparency and reproducibility of the study.

Response: Yes, to enhance the transparency and reproducibility of our frailty classification methodology, we have now added a detailed description of the parameters, criteria, and thresholds used in the frailty index. Please refer to the supplementary materials for further details. We have revised this part, highlighted in red in the manuscript, and the revised text is transcribed below:

“The frailty index comprised 49 deficits spanning multiple systems [45], including Cognition (Experience confusion/memory problems), Dependence(Managing money, Stooping, crouching, kneeling, and Lifting or carrying; etc., 15 items), Depressive Symptoms (Have little interest in doing things, Trouble sleeping or sleeping too much and Feeling tired or having little energy; etc., 7 items), Comorbidities (Arthritis, Thyroid problems and Chronic bronchitis; etc., 13 items), Hospital Utilization and Access to Care (Self-rated health, Health now compared with 1 year ago and Overnight hospital patient in past year; etc., 6 items), Laboratory Values (Glycohemoglobin, Red blood cell count and Hemoglobin; etc., 6 items) (Table S1).” In this study, fit and vulnerable (i.e., FI≤0.2) were combined into the no-frailty group, while mildly frail and moderately/severely frail (i.e., FI > 0.2) were combined into the frailty group.

Supplementary Table 1

Variables in the 49-Item Frailty Index and Their Respective Scorings.

Q3: I suggest that the authors specify which sampling weight was used to ensure the representativeness of the data, given that the study is based on a complex sampling design. Including this information would enhance the clarity and validity of the statistical approach.

Response: We sincerely apologize for not incorporating weighting in our original statistical analysis, and we fully understand the importance of this information, especially given the complex sampling design of our study. Prior to conducting our research, we reviewed numerous studies, many of which also did not apply weighting [1-6]. However, in response to your feedback, we have now incorporated sampling weights into our analysis. Detailed information on the weighting method, including the specific weights applied, has been added to the sensitivity analysis section of the manuscript. Additionally, the results of the weighted analysis have been included in the supplementary materials, where the weighting process is further elaborated. The inclusion of sampling weights was carefully considered, and we have verified that it does not significantly alter the overall conclusions of the study. Nevertheless, it does provide a more robust and representative analysis of the data. We have revised this part, and highlighted in red in the manuscript, and the revised text is transcribed below:

Statistical analysis

“…. The mediation package in R 4.3.3 was used to examine the mediating association of serum 25(OH)D3 levels between dietary inflammation and frailty. Numerous studies have utilized publicly available data from the NHANES to investigate risk factors for various diseases. In these studies, some researchers have employed weighted analysis methods, while others have used unweighted approaches. Although NHANES employs complex sampling techniques to enhance the representativeness and generalizability of the survey results, discrepancies in conclusions may arise between weighted and unweighted analyses. In the present study, we conducted a sensitivity analysis using weighted logistic regression to reaffirm our findings. The weighting variable wtdrd2 were selected, and the combined weight was calculated as 1/2*wtdrd2. All sensitivity analyses were conducted using these weighted approaches.”

Sensitivity analysis

Similarly, the sensitivity analysis using weighted logistic regression indicated that when DII was treated as a continuous variable, a one-unit increase in DII was associated with a 9% increase in the risk of frailty in Model 3 (OR = 1.09, 95% CI: 1.01–1.18, P = 0.019). When DII was dichotomized (pro-inflammatory vs. anti-inflammatory), individuals with a pro-inflammatory diet had an 18% higher risk of frailty compared to those with an anti-inflammatory diet in Model 3 (OR = 1.18, 95% CI: 0.89–1.57, P = 0.216), although this association was no longer statistically significant. When DII was categorized into quartiles, individuals in the highest quartile (Q4) had a 62% higher risk of frailty compared to those in the lowest quartile (Q1) in Model 3 (OR = 1.62, 95% CI: 1.16–2.26, P = 0.006). (Table 6). The weighted mediation effect accounted for 6.8% of the total variance (0.0129/0.1886). (Table 7)

[1] Liu H, Tan X, Liu Z, Ma X, Zheng Y, Zhu B, Zheng G, Hu Y, Fang L, Hong G. Association Between Diet-Related Inflammation and COPD: Findings From NHANES III. Front Nutr. 2021 Oct 18;8:732099. doi: 10.3389/fnut.2021.732099. PMID: 34733875; PMCID: PMC8558221.

[2] Liu Z, Liu H, Deng Q, Sun C, He W, Zheng W, Tang R, Li W, Xie Q. Association Between Dietary Inflammatory Index and Heart Failure: Results From NHANES (1999-2018). Front Cardiovasc Med. 2021 Jul 6;8:702489. doi: 10.3389/fcvm.2021.702489. PMID: 34307508; PMCID: PMC8292138.

[3] Wu L, Shi Y, Kong C, Zhang J, Chen S. Dietary Inflammatory Index and Its Association with the Prevalence of Coronary Heart Disease among 45,306 US Adults. Nutrients. 2022 Oct 28;14(21):4553. doi: 10.3390/nu14214553. PMID: 36364813; PMCID: PMC9656485.

[4] Sun L, Huo X, Jia S, Chen X. The Association betw

---

## [Decision Letter · Decision Letter 1]

PONE-D-24-52566R1The serum vitamin D levels alleviate the influence of dietary inflammation on frailty: a cross-sectional analysis in the U.S. older adultsPLOS ONE

Dear Dr. Wang,

Thank you for submitting your manuscript to PLOS ONE. After careful consideration, we feel that it has merit but does not fully meet PLOS ONE’s publication criteria as it currently stands. Therefore, we invite you to submit a revised version of the manuscript that addresses the points raised during the review process.

We look forward to receiving your revised manuscript.

Kind regards,

Everson Nunes, Ph.D.

Academic Editor

PLOS ONE

Journal Requirements:

**Additional Editor Comments:**

Dear Authors,

Thank you for all your work on improving the manuscript and assessing the reviewers' comments and questions. I am please to accept the manuscript once one very minor thing pointed by one of the reviewers is fixed. Please check reviewer 3 comments for that.

Reviewers' comments:

Reviewer's Responses to Questions

**Comments to the Author**

1. If the authors have adequately addressed your comments raised in a previous round of review and you feel that this manuscript is now acceptable for publication, you may indicate that here to bypass the “Comments to the Author” section, enter your conflict of interest statement in the “Confidential to Editor” section, and submit your "Accept" recommendation.

Reviewer #1: (No Response)

Reviewer #3: (No Response)

2. Is the manuscript technically sound, and do the data support the conclusions?

Reviewer #1: (No Response)

Reviewer #3: Yes

3. Has the statistical analysis been performed appropriately and rigorously? 

Reviewer #1: (No Response)

Reviewer #3: Yes

4. Have the authors made all data underlying the findings in their manuscript fully available?

Reviewer #1: (No Response)

Reviewer #3: Yes

5. Is the manuscript presented in an intelligible fashion and written in standard English?

Reviewer #1: (No Response)

Reviewer #3: Yes

6. Review Comments to the Author

Reviewer #1: (No Response)

Reviewer #3: Dear authors

- "Ratio of family income to poverty (RIP)" should be changed to "Ratio of family income to poverty (PIR)". In some places, still "RIP" is used for ratio of family income to poverty. Check throughout the manuscript.

7. PLOS authors have the option to publish the peer review history of their article (what does this mean? ). If published, this will include your full peer review and any attached files.

**Do you want your identity to be public for this peer review?** For information about this choice, including consent withdrawal, please see our Privacy Policy .

Reviewer #1: No

Reviewer #3: No

---

## [Author Response · Author response to Decision Letter 2]

7 Jun 2025

Dear Editors and reviewers,

We sincerely appreciate your time and valuable comments, which have helped improve our manuscript. The submission ID is PONE-D-24-52566R2.

We hereby submit our responses to the reviewers' comments. We hope the revised manuscript now meets the publication standards of PLOS ONE.

Q1: "Ratio of family income to poverty (RIP)" should be changed to "Ratio of family income to poverty (PIR)". In some places, still "RIP" is used for ratio of family income to poverty. Check throughout the manuscript.

Response: We sincerely appreciate the reviewer's careful reading. We have systematically replaced all instances of "Ratio of family income to poverty (RIP)" with the standard abbreviation "PIR" throughout the manuscript. We have made revisions to this part, highlighted in red in the manuscript, and the revised text is transcribed below:

Line 162: “….model 2 adjusting for sex, age, education, (RIP) PIR and ….”

Line 187: “….characteristics (age, sex, ethnicity, marriage, (RIP) PIR, clinical….”

---

## [Editor Report · Decision Letter 2]

The serum vitamin D levels alleviate the influence of dietary inflammation on frailty: a cross-sectional analysis in the U.S. older adults

PONE-D-24-52566R2

Dear Dr. Wang,

We’re pleased to inform you that your manuscript has been judged scientifically suitable for publication and will be formally accepted for publication once it meets all outstanding technical requirements.

Kind regards,

Everson Nunes, Ph.D.

Academic Editor

PLOS ONE
---

## [Editor Report · Acceptance letter]

PONE-D-24-52566R2

PLOS ONE

Dear Dr. Wang,

I'm pleased to inform you that your manuscript has been deemed suitable for publication in PLOS ONE. Congratulations! Your manuscript is now being handed over to our production team.

Kind regards,

on behalf of

Dr. Everson Nunes

Academic Editor

PLOS ONE